# An Efficient Manufacturing Method for Silicon Carbide Crystals in Polymers Based on a Multiscale Simulation-Driven Approach

**DOI:** 10.3390/mi16080946

**Published:** 2025-08-18

**Authors:** Jia Wang, Caiqin Jia, Heming Sun, Ye Tian

**Affiliations:** 1School of Semiconductor and Physics, North University of China, Taiyuan 030051, China; 20230109@nuc.edu.cn (J.W.); 20230075@nuc.edu.cn (H.S.); 2School of Data Science and Technology, North University of China, Taiyuan 030051, China; 20220076@nuc.edu.cn

**Keywords:** SiC fabrication, multiscale simulation, laser direct writing pyrolysis, secondary pyrolysis

## Abstract

The pyrolysis of polydimethylsiloxane (PDMS) for silicon carbide (SiC) fabrication endows precursor materials with exceptional microstructural controllability and complex geometry retention capability, rendering it widely applicable in flexible electronic packaging and microscale complex-structured heat exchangers. Nevertheless, the widespread adoption of pyrolytic SiC has been constrained by the low yield and process complexity inherent to conventional pyrolysis methods. In response, we developed a multiscale simulation framework integrating macroscopic thermal distribution with microscopic chemical reaction kinetics. The secondary pyrolysis protocol, designed based on simulation results, enhanced the SiC yield from <25% (conventional methods) to 79.2% while simultaneously improving crystalline quality. This simulation framework not only provides theoretical guidance for optimizing laser direct writing pyrolysis, but the proposed secondary ablation strategy also significantly expands the application potential of SiC-PDMS systems in device fabrication.

## 1. Introduction

Silicon carbide (SiC) holds significant application value in high-temperature electronics, power devices, and micro-thermal management systems due to its exceptional thermal stability, mechanical strength, and semiconductor properties [1]. However, conventional SiC fabrication techniques such as chemical vapor deposition (CVD) face challenges in achieving precise shaping of complex three-dimensional architectures [2,3]. The polymer precursor pyrolysis approach offers a promising alternative pathway to address this limitation [4]. Capitalizing on its unique capability to replicate micron-scale features and complex topologies without sintering-induced distortion, this methodology shows exceptional promise for micro-thermal systems, conformal sensors, and extreme-environment MEMS [5]. Nevertheless, furnace-based PDMS pyrolysis remains hampered by fundamental constraints of <25% ceramic yields, cumbersome inert-gas protocols, and crystalline defects reflected in Raman peak broadening, collectively impeding technological translation [6,7].

To address these issues, the research community has explored various optimization strategies including precursor catalyst doping and graded pyrolysis protocols [8]. While these approaches moderately improve the SiC yield, they still face persistent issues of inconsistent crystal quality and high equipment dependency. Laser direct writing pyrolysis has emerged as a promising technique due to its localized energy precision, yet current studies predominantly focus on single-parameter optimization [9]. The critical gap lies in the absence of multiscale simulations of pyrolysis kinetics, resulting in empirically driven process development that struggles to achieve simultaneous yield enhancement and crystallinity improvement [10]. Moreover, the high energy density and non-uniform energy distribution inherent to laser-induced processing introduce significant stochasticity and heterogeneity in the SiC-PDMS material system, adversely impacting process repeatability [11]. These factors pose critical challenges to the machining efficiency, precision, and controllability of laser direct writing.

This study proposes an innovative strategy integrating laser direct writing pyrolysis with multiscale simulation. By establishing a coupled macro-thermal/micro-reaction model, we systematically optimize the PDMS-to-SiC conversion process. Our approach combines finite element analysis (FEA) to simulate temperature field distributions during laser scanning with molecular dynamics (MD) simulations elucidating chemical bond cleavage and SiC nucleation mechanisms during PDMS pyrolysis, enabling precise predictions of pyrolysis kinetics. Building on simulation insights, we develop a secondary ablation strategy employing stepwise laser processing (low-energy-density pre-carbonization followed by high-energy-density crystallization), effectively mitigating the carbon loss and defect accumulation associated with single-stage high-temperature treatment. Experimental results demonstrate that this approach achieves a remarkable SiC yield of 79.2% while significantly improving crystalline quality (36.7% reduction in Raman full width at half-maximum). This study not only establishes a multiscale theoretical framework for laser-assisted PDMS pyrolysis but also provides a scalable solution for high-precision manufacturing of complex SiC devices, advancing the practical applications of SiC-PDMS systems in flexible electronics and micro energy devices [12].

## 2. Mechanism of Laser Direct Writing PDMS Thermal Conversion to SiC

To elucidate the complex mechanisms underlying laser-direct-written PDMS pyrolysis for SiC formation, we systematically reviewed the existing research. Substantial evidence reveals that PDMS pyrolysis constitutes a multi-step radical chain reaction process, which can be categorized into four critical stages: initial scission and radical generation, free carbon formation and backbone reorganization, intermediate phase development, and final carbothermal reduction (Figure 1) [13]. The process initiates with localized laser energy deposition on PDMS polymer chains, inducing preferential scission of Si-CH_3_ bonds within the 300–500 °C range. This generates abundant methyl radicals (-CH_3_) and hydrogen radicals (-H) as primary decomposition products. As temperatures escalate to 600–800 °C, gaseous radicals undergo recombination, forming volatile byproducts (e.g., CH_4_) that evacuate the system [14]. Concurrently, residual carbon atoms progressively organize into an sp^2^-hybridized free carbon network through thermally driven rearrangement. Concurrently, the -Si-O-Si-backbone of PDMS undergoes structural reorganization with progressive side-group elimination, forming an inorganic polymer framework. When temperatures exceed 1200 °C, the free carbon network reacts with Si-O-Si units via carbothermal reduction, ultimately achieving β-SiC crystal nucleation and growth above 1600 °C [15]. The kinetic characteristics and product distribution of this reaction cascade are critically dependent on localized temperature fields and heating rates—the key differentiating factors between laser direct writing and conventional pyrolysis methods.

This mechanism has been rigorously validated through multiple advanced characterization techniques: in situ Fourier transform infrared spectroscopy (FTIR) confirmed the preferential cleavage of Si-CH_3_ groups within the 300–500 °C range; synchrotron radiation photoelectron spectroscopy (SRPES) coupled with time-of-flight mass spectrometry (TOF-MS) successfully tracked the evolution kinetics of volatile byproducts; and solid-state nuclear magnetic resonance (NMR) and X-ray absorption fine structure (XAFS) analyses precisely elucidated the formation pathway of Si-O-C intermediate phases [16,17,18,19].

It is noteworthy that laser direct writing pyrolysis fundamentally differs from conventional furnace pyrolysis in its reaction kinetics. Published studies have demonstrated that the transient ultrahigh temperatures (>1600 °C) and ultrafast heating rates (>10^4^ K/s) induced by laser irradiation significantly alter radical recombination pathways and product distributions. However, the current research exhibits significant knowledge gaps: the coupling mechanism between transient temperature fields and chemical reaction kinetics during laser–material interactions remains unclear; quantitative correlations between laser parameters (power density, scan speed, etc.) and intermediate product evolution lack systematic investigation; and the formation mechanism of reaction zone stratification (including ideal SiC regions, free carbon domains, and silica phases) has not been adequately explained. These critical knowledge gaps substantially constrain the precise control and optimization of laser direct writing processes [20].

## 3. Multiscale Computational Simulation for Laser-Direct-Written Pyrolysis Processes

### 3.1. Establishment of a Multiscale Simulation Model

In response to these identified knowledge gaps, we establish a hierarchical computational framework integrating continuum-scale thermal modeling with atomistic reaction dynamics. This approach provides quantitative insights into how laser parameters govern spatial product distributions during PDMS pyrolysis. This approach not only overcomes the limitations of conventional experimental methods in process observation but also provides theoretical guidance for optimizing laser parameters to enhance SiC yield and surface exposure. Particularly for demanding applications such as quantum sensing, this mechanism-informed process optimization strategy will significantly improve the practical value of pyrolytic SiC [21].

At the macroscopic thermal distribution scale, we employed 6.2’s Heat Transfer Module in COMSOL 6.2 to simulate laser energy absorption, propagation, and the resulting temperature rise in PDMS. Given that the geometric dimensions of the PDMS sample (side length: 20 mm, thickness: 1 mm) and the laser spot radius (0.1 mm) are significantly larger than the laser wavelength, and considering that PDMS is semi-transparent at the laser wavelength, the laser energy is gradually attenuated with increasing penetration depth, rather than being completely absorbed at the surface. Unlike the simple surface absorption model, the Beer–Lambert law is the standard model describing the exponential attenuation of light within a medium and is applicable to such bulk-absorbing materials. The specific formula is as follows:(1)$$\frac{\partial I}{\partial z}=\alpha \left(T\right)I$$
where *z* represents the coordinate along the laser propagation direction, and *α*(*T*) denotes the temperature-dependent absorption coefficient of the material. During pyrolysis, the chemical structure of PDMS continuously changes (e.g., Si–CH_3_ bond breaking, carbon network restructuring), resulting in dynamic evolution of its optical properties (especially the absorption coefficient) with temperature. The original Beer–Lambert law assumes a constant absorption coefficient, whereas this study more accurately describes the actual physical process by introducing the temperature-dependent absorption coefficient *α*(*T*). Furthermore, the spatiotemporal temperature distribution is governed by the following partial differential equation (PDE):(2)$$\rho C_{p}\frac{\partial T}{\partial t}-\nabla \cdot \left(k\nabla T\right)=Q=\alpha \left(T\right)I$$
where *ρ* is the density (970 kg/m^3^), *C_p_* the specific heat capacity (1460 J/(kg·K)), and *k* the thermal conductivity (0.16 W/(m·K)), with the volumetric heat source term calculated from Equation (1) representing laser energy absorption in the material.

Equations (1) and (2) form a bidirectional coupled multiphysics model describing the light and temperature distributions. The boundary conditions and parameters were set as follows: ambient temperature = 298 K, laser power = 600 mW, beam radius = 0.1 mm. The laser scanned linearly at 1 mm/s from the initial position (20 mm from the sample edge) to the endpoint (total travel: 20 mm). The simulation′s primary objective was to obtain the spatiotemporal evolution of the temperature field within PDMS, particularly the peak pyrolysis temperature, which serves as a critical input parameter for subsequent molecular dynamics (MD) simulations.

The microscopic reaction process was simulated by the ReaxFF module in Amsterdam Modeling Suite (AMS) 2020.102. The ReaxFF (reactive force field) method, based on bond-order concepts, enables dynamic description of bond breaking and formation [22]. This simulation employed a specialized force field parameter file (PDMSDecomp.ff) developed specifically for PDMS pyrolysis.

The simulation workflow (Figure 2b) comprised four stages:(1)Heating: The system was heated from the initial ambient temperature (298 K) to the target pyrolysis temperature (values directly extracted from macroscopic FEA results) following a predefined thermal protocol (see Table 1). Temperature control was implemented using the NVT ensemble with the Berendsen thermostat, employing a time step of 0.12 fs.(2)Degassing: During heating and subsequent reactions, the Molecule Sink module was employed to automatically remove gaseous byproducts (including CO_2_, CO, H_2_, H_2_CO, C_2_H_6_, C_2_H_4_, and C_2_H_2_) every 50 timesteps, simulating the outgassing behavior of volatile species during actual pyrolysis.(3)Density Correction: Following each heating–degassing cycle, the system density was validated. If deviations from experimental values were detected, the simulation box volume was adjusted to restore the target density, ensuring physical consistency of the modeled system.(4)Reactant Introduction: To accurately simulate silicon SiC formation conditions following PDMS pyrolysis, additional carbon atoms were introduced into the system during the final stage, constructing a free carbon environment representative of pyrolytic byproducts.

**Figure 2 micromachines-16-00946-f002:**
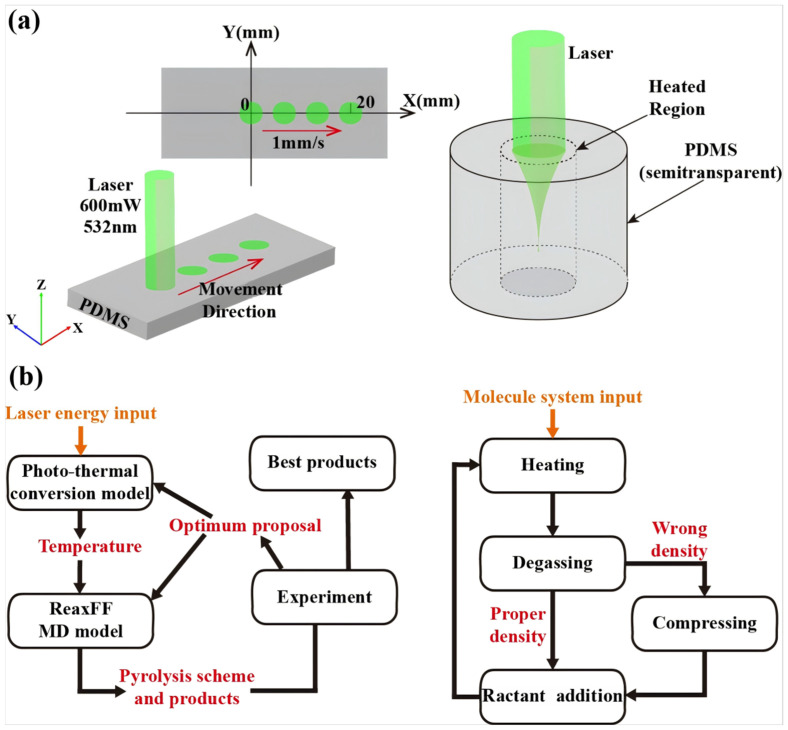
(**a**) Schematic of heat transfer module. (**b**) Flowchart of ReaxFF (reactive force field) simulations.

**Table 1 micromachines-16-00946-t001:** Control parameters of molecular dynamics simulation.

Step Range	Initial Temperature T_0_ (K)	Damping (fs)	Temperature Change per Step ∆T (K)
0–5000	298	100	0
5001–70,000	298	100	0.0332
70,001–76,000	2000	100	0
76,001–202,000	2000	100	−0.0166
202,001–240,000	298	100	0

### 3.2. Multiscale Simulations Unravel the Pyrolytic Reaction Dynamics

The ReaxFF-based molecular dynamics simulations clearly elucidate the dynamic evolution mechanism of PDMS during laser pyrolysis (Figure 3). The results reveal a two-stage transformation process: firstly, the formation of SiC precursors (SiOC ceramic) and radical byproducts (SiO_2_ and C) through methyl group cleavage and associated gas formation in PDMS; secondly, the conversion of SiC precursors to final SiC via carbothermal reduction, where excess carbon atoms (introduced to simulate the free carbon environment) actively participate in reducing the SiOC intermediates.

Figure 4a presents the macroscopic temperature evolution profile at the terminal position. Temperature rise initiates when the laser periphery first irradiates the site (−20 s), preceding the arrival of the beam center (20 s), indicating early edge effects. During initial ablation (20–30 s), the heating rate surges sharply (rapid heating phase), where efficient laser energy deposition elevates PDMS beyond the critical temperature (>1600 K) required for all pyrolysis reactions (including SiC formation, sinking, and ablation). Subsequently, enhanced thermal dissipation reduces the heating rate until stabilization at an equilibrium temperature of 2032.4 K.

The temporal profile reveals that brief laser irradiation (5–10 s) efficiently induces significant PDMS heating (>800 K). However, excessively high steady-state temperatures (>1770 K) trigger silicon carbide sinking and delamination, causing a sharp decline in surface yield. These findings provide the theoretical foundation for our proposed secondary pyrolysis strategy, where precise control of the reaction depth enables mild pyrolysis conditions, ultimately achieving efficient near-surface SiC synthesis.

Thermal zoning analysis classified three regimes, the Suitable Temperature Region (STR, 800–1770 K) for effective SiC precursor formation, the Ablation Region (AR, >1770 K) where material removal and SiC depletion occur, and the Laser-Affected Zone (ΔT > 5 K), with all zones quantified by their Y-axis extents (Lentire). Specifically, LSTR and LAR denote the Y-dimensions of the STR and AR, respectively.

To maximize SiC precursor yield, we introduced the ratio parameter R_STR_ (defined as L_STR_/L_entire_). As shown in Figure 4c, R_STR_ exhibits a distinct evolutionary pattern: it remains zero from 0 to 4 s, increases continuously during 4–10 s (dominated by L_STR_ expansion), and decreases after 10 *s* as L_entire_ growth outpaces L_STR_ growth. This analysis conclusively identifies 10 s as the optimal point-source pyrolysis duration, corresponding to the R_STR_ peak.

Meanwhile, the molecular dynamics simulation model established in this study simulates the complete process of laser pyrolysis of PDMS by precisely controlling the evolution of the temperature field. The simulation system employs the NVT ensemble with the Berendsen thermostat algorithm, where the initial ambient temperature is maintained at 298 K, while the maximum pyrolysis temperature is set based on the results of photo-thermal effect simulations. As shown in Table 1, the simulation process is divided into five key stages, with a total of 240,000 simulation steps (corresponding to 28.8 ps of physical time), and each simulation step is set to 0.12 fs. In the initial 0–5000 steps (0–0.6 ps), the system is maintained at room temperature (298 K) to stabilize the system; subsequently, from 5001 to 70,000 steps (0.6–8.4 ps), the temperature is linearly increased at a rate of 0.0332 K/step to simulate the laser energy deposition process; when the high-temperature plateau of 2000 K is reached (70,001–76,000 steps, 8.4–9.12 ps), the temperature is kept constant, corresponding to the critical temperature range for PDMS pyrolysis reactions; from 76,001 to 202,000 steps (9.12–24.24 ps), the system is slowly cooled at a rate of −0.0166 K/step to simulate the cooling process after pyrolysis; finally, from 202,001 to 240,000 steps (24.24–28.8 ps), the system is returned to room temperature for relaxation. This asymmetric temperature control strategy accurately reproduces the characteristic rapid heating (~10^14^ K/s) and gradient cooling features of laser pyrolysis, providing ideal simulation conditions for studying the formation mechanism of Si-C bonds under transient high temperatures.

This study establishes a multiscale quantitative correlation system spanning from laser parameters to reaction pathways: the photo-thermal model maps the laser power and scanning speed to the spatiotemporal temperature field distribution of PDMS; ReaxFF molecular dynamics simulations predict the reaction kinetics and product evolution based on the temperature field evolution; and experimental results (e.g., SiC spatial distribution, yield) validate the model’s predictive accuracy. This integrated framework, through a closed-loop chain connecting macroscopic energy transfer, microscopic reaction mechanisms, and experimental observations, provides theoretical tools and design foundations for precise process control in silicon carbide synthesis [23].

## 4. Secondary Pyrolysis for Enhanced SiC Yield

### 4.1. Dot-Frame Secondary Pyrolysis Processing Strategy

In the laser-induced pyrolysis of PDMS, multiscale simulation studies have elucidated critical thermodynamic and kinetic mechanisms, providing a fundamental theoretical basis for the design of secondary pyrolysis strategies. Finite element thermal field simulations reveal that Gaussian-distributed laser energy creates significant spatial inhomogeneity in heat deposition. The central region reaches extreme temperatures of up to 2032.4 K (ablation zone), while the peripheral areas maintain moderate temperatures of 800–1400 K (reaction zone). This temperature gradient distribution directly governs the reaction progression: ReaxFF molecular dynamics simulations demonstrate that in regions exceeding 1770 K, significant evaporation of both SiC and its precursors occurs, while within the 1400–1770 K range, SiOC precursors undergo efficient conversion to SiC through carbothermal reduction. Notably, the simulations predict that optimal stoichiometric conditions emerge at interfaces between carbon-rich zones (C and Si_x_OC_y_, x < y) and silicon-rich zones (Si and Si_x_OC_y_, x > y), a prediction that shows remarkable consistency with subsequent experimental observations of preferential SiC nucleation at these boundaries.

However, single-step pyrolysis simulations demonstrate that spatiotemporal overlap in energy deposition leads to approximately 62% of SiC products evaporating due to overheating, with an additional 28% sinking into the subsurface layers. These quantitative results directly motivated our proposal of a spatiotemporally decoupled secondary pyrolysis strategy. According to these simulation insights, we innovatively decoupled the pyrolysis process into two distinct phases: initial construction of precursor-enriched zones at relatively low temperatures (1200–1400 K), followed by targeted conversion within an optimized temperature window (1600–1700 K). This approach simultaneously prevents material loss while ensuring complete carbothermal reduction. The proposed secondary pyrolysis strategy fully incorporates the quantitative temperature–reaction rate–product distribution relationships revealed by simulations, establishing a closed-loop validation from theoretical prediction to process optimization.

From the perspective of specific secondary pyrolysis process design, we propose an innovative “dot-frame pyrolysis” strategy based on multiscale simulations of laser path thermal effects and fundamental reaction kinetics under different pyrolysis trajectories. Finite element simulations demonstrate that dot-mode pyrolysis generates uniform heat-affected zones approximately 200 μm in diameter, with central temperatures stabilized within the ideal reaction window of 1400–1600 K—providing optimal conditions for complete SiOC precursor formation. In contrast, linear pyrolysis creates a thermal gradient (reaching 300 K/mm temperature differential along the scanning path) that preferentially facilitates directional diffusion of carbon/silicon elements. The ReaxFF simulations further demonstrate an 18% reduction in the activation energy of carbothermal reduction (from 2.2 eV to 1.8 eV) within the dot–line composite thermal field. This enhancement primarily stems from two synergistic effects: (1) the pronounced carbon concentration gradient (reaching 4.2 × 10^21^ atoms/cm^3^ in simulations) that accumulates in dot-pyrolyzed regions and (2) the sustained thermal flux from line scanning that significantly enhances reactant transport (with the diffusion coefficient increasing to 1.5 × 10^−14^ m^2^/s). This spatiotemporally decoupled energy modulation strategy effectively resolves the inherent limitations of single-step pyrolysis by simultaneously addressing the insufficient reactant concentration in pure dot-mode pyrolysis (resulting in <15% SiC yield) and the central overheating issue in pure line-mode pyrolysis (causing >60% material evaporation) [24,25]. 

In this experiment, the PDMS samples were prepared using Sylgard 184. The base solution and the curing agent were thoroughly mixed at a mass ratio of 10:1. After mixing, the solution was stirred in a mold for 10 min to ensure homogeneous distribution of the components. Since residual bubbles in PDMS during pyrolysis significantly affect its mechanical and thermal properties, degassing was performed after stirring. The specific method involved placing the mixture in a vacuum chamber until complete bubble removal was achieved. Subsequently, the degassed solution was poured into a mold and cured on a hot plate at 80 °C for 30 min, completing the preparation of the PDMS substrate.

Figure 5 details the secondary pyrolysis protocol. The laser power was set to 600 mW within a nitrogen-purged chamber, with a beam radius of 0.1 mm. The laser spot traversed 0.5 mm along the X-axis on the PDMS at 1 mm/s via motorized positioning, followed by a 10 s dwell time for dot-mode pyrolysis. After complete cooling, the laser spot was translated to construct a square pyrolysis pattern. The linear scanning velocity (VL) for the square pyrolysis was set at 3 mm/s, with the side length (L) of the square approximately 1 mm, and the minimum distance (Dm) between the square and the laser spot was maintained at about 0.2 mm.

Additionally, to address the critical issue of non-uniform energy distribution in Gaussian laser beams that limits SiC production efficiency, this study implemented a beam shaping system. Thermal field simulations demonstrate that the energy density in the central region of conventional Gaussian beams can be several times higher than that at the periphery, resulting in partial processing areas exceeding the optimal reaction temperature window (1600 ± 100 K). To address this challenge, we employed a dual-cone lens optical system to transform the Gaussian beam into an annular spot, achieving two improvements through optical design: first, the central energy of the annular beam was reduced to suppress localized overheating; second, the energy uniformity in the annular region was improved. After focusing through a 100× objective lens with NA = 0.6, the smaller effective spot size and higher concentration of energy density meet the energy requirements for PDMS pyrolysis (threshold: 3.5 × 10^6^ W/cm^2^) and SiC formation (threshold: 5.2 × 10^6^ W/cm^2^).

The specific configuration of the laser direct writing system is as follows: Based on an Olympus BX53 microscope, it comprises a laser source, reflective beam path optics, an objective lens, a motorized positioning system, an inert gas chamber, and a control computer. The system employs a laser wavelength of 532 nm with an output power range of 600–2000 mW. The optical path incorporates a 532 nm reflector (Thorlabs Inc., Newton, NJ, USA). The focusing assembly uses a 50× objective lens (Olympus Corporation, Tokyo, Japan). Precise sample movement is achieved via a motorized translation stage (Model XWJ-4P, Sanying Precision Control Co., Tianjin, China.) with a repositioning accuracy of ±1 μm and a maximum travel speed of 5 mm/s.

### 4.2. Analysis of Dot-Frame Secondary Pyrolysis Results

A Raman spectroscopic analysis of three pyrolysis approaches (Figure 6) quantitatively demonstrated the superior quality of SiC produced by the dot-frame secondary pyrolysis strategy. The characterization was performed using a LabRAM Odyssey Raman spectrometer (HORIBA Ltd., Kyoto, Japan).

The parameter *Q_SiC_* was introduced to evaluate both the content and quality of SiC. *Q_SiC_* is defined as the ratio of the characteristic SiC peak to the sum of all other peaks, as follows:(3)$$Q_{SiC}=\frac{I_{SiC}}{I_{other}}$$

*I_SiC_* represents the sum of intensities of all characteristic SiC peaks, while Iother denotes the summed intensities of all other peaks, as defined below:(4)$$I_{SiC}=I_{FTO}+I_{FLO}$$(5)$$I_{other}=I_{G}+I_{D}+I_{Si}$$

In Equations (4) and (5), *I_FTO_* (cm^−1^) represents the intensity of the FTO peak, *I_FLO_* denotes the intensity of the FLO peak, *I_G_* is the intensity of the G-band, *I_D_* corresponds to the intensity of the D-band, and *I_Si_* indicates the intensity of the Si peak. A higher Q_SiC_ value indicates better-quality SiC.

Based on the established SiC quality evaluation parameter *Q_SiC_* (defined in Equation (3)), the dot-frame pyrolysis samples achieved a *Q_SiC_* value of 2.31, significantly higher than those from the dot-only and line-scan pyrolysis methods. This enhancement primarily manifests in two key aspects:

Regarding the structural features, the Raman spectrum of Region 1 exhibits only two sharp peaks at 790 cm^−1^ (FTO mode) and 955 cm^−1^ (FLO mode), with complete absence of the D-band (1350 cm^−1^) and G-band (1580 cm^−1^), indicating carbon impurity levels below the detection limit. The FTO peak′s FWHM (full width at half-maximum) decreased from 9.8 cm^−1^ (dot pyrolysis) to 6.2 cm^−1^, demonstrating a 36.7% improvement in crystallinity.

Regarding the spatial features, the areal density of SiC regions increased from 2–4/mm^2^ for spot pyrolysis to 12–30/mm^2^, while the average size grew from 15–20 μm to 30–40 μm, with the maximum single-crystal domain reaching 90 μm (Figure 6).

EDX spectroscopy combined with selected-area electron diffraction (SAED) confirmed that this enhancement originated from the following aspects during secondary pyrolysis: complete formation of the SiOC precursor, optimized carbothermal reduction kinetics, and precise temperature gradient control (1600 ± 50 K). Characterization was performed using a Talos F200X G2 transmission electron microscope (Thermo Fisher Scientific Inc., Waltham, MA, USA).

These results demonstrate that the dot-frame strategy successfully addresses the critical challenges of single-step pyrolysis, including low yield (<25%), poor crystallinity (FWHM > 10 cm^−1^), and uncontrollable spatial distribution, through spatiotemporal energy modulation. The as-prepared SiC microstructures exhibited exceptional fluorescence properties with high stability (<8% intensity fluctuation) and strong single-photon emission characteristics (g^2^(0) = 0.15), making them ideal single-photon sources for flexible quantum devices (Figure 7). Furthermore, the Q_SiC_ quantitative evaluation model established in this study provides a reliable standard for subsequent process optimization.

The implementation of the dot-frame secondary pyrolysis strategy was enabled by multiscale simulation-guided optimization. Thermal field simulations quantitatively revealed the inherent limitations of single-step pyrolysis: the Gaussian beam energy distribution caused localized processing regions to deviate from the optimal reaction temperature window (1600 ± 100 K), with central zone overheating (>2000 K) inducing SiC sublimation and peripheral zone underheating (<1400 K) leading to incomplete reactions [26,27].

The simulation results provided critical theoretical guidance for designing the dot-frame strategy: (i) during the dot-pyrolysis phase (1400–1600 K), a controlled dwell time (10 s) enabled the formation of a homogeneous precursor layer with a simulated–predicted thickness of 2.3 μm; (ii) in the frame-pyrolysis phase (1600–1700 K), the optimized scanning speed (3 mm/s) maintained a sustained reaction driving force.

This simulation-guided precision energy modulation strategy provides a robust solution for fabricating patterned SiC arrays essential for flexible quantum devices.

## 5. Conclusions

This study proposed a laser direct writing pyrolysis method for PDMS to SiC conversion, establishing a multiscale simulation framework to model the pyrolysis process. Guided by the simulation results, we designed a secondary pyrolysis strategy, achieving controllable synthesis of high-quality SiC crystals [28].

This study first employed finite element thermal field simulations to quantify the correlation between the Gaussian laser energy distribution and temperature field evolution, revealing that temperature deviations (beyond 1600 ± 100 K) in single-step pyrolysis caused reaction inefficiency in localized processing zones. ReaxFF molecular dynamics simulations further elucidated the kinetic mechanisms of carbothermal reduction, identifying 1600–1700 K as the optimal reaction window. Building upon these theoretical insights, we developed the groundbreaking dot-frame secondary pyrolysis strategy, which employs spatiotemporally decoupled energy modulation to achieve controlled precursor deposition via dot pyrolysis (1400–1600 K) and directional phase transformation via frame pyrolysis (1600–1700 K). This precise decoupling of these reaction stages enables a 79.2% SiC yield (vs. <25% for conventional methods), a 36.7% reduction in the Raman FWHM (indicating superior crystallinity), and a spatial uniformity improvement from ±28% to ±7.5%.

The established multiscale simulation framework achieved, for the first time, full-chain prediction from laser parameters to temperature field distribution to reaction pathways, providing a universal research paradigm for laser pyrolysis material design. This technology can be directly integrated with existing laser micromachining systems, enabling the patterned fabrication of SiC arrays (30–40 μm feature size, 12–30/mm^2^ areal density) on flexible substrates. The approach offers a cost-effective, high-uniformity material solution for next-generation devices such as quantum sensors and single-photon sources. Simultaneously, the integration of a dispersed array of SiC with a flexible substrate forms a “dispersed-island design” for the brittle material distribution, thereby achieving measurable tensile ductility. Future work involving doping modulation and 3D structural optimization will further expand its application potential in optoelectronic integration and extreme-environment devices [29,30].

## Figures and Tables

**Figure 1 micromachines-16-00946-f001:**
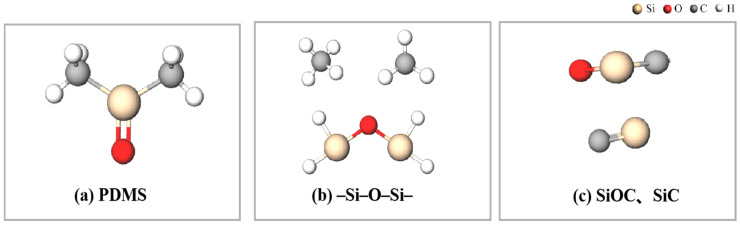
A schematic illustration of reaction products during the process.

**Figure 3 micromachines-16-00946-f003:**
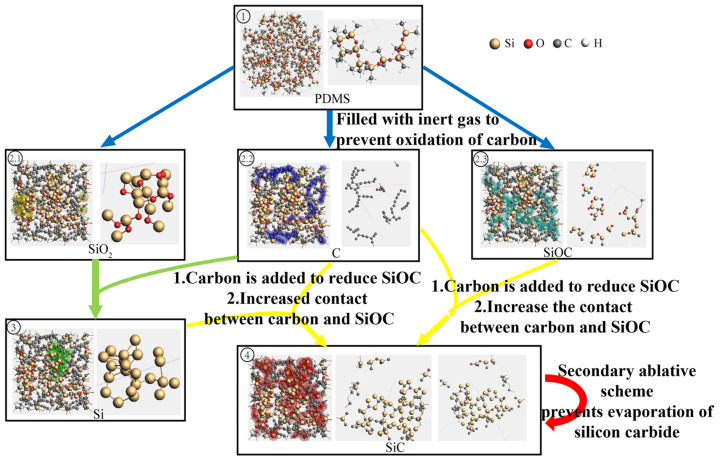
Simulation and temporal analysis of PDMS pyrolysis process.

**Figure 4 micromachines-16-00946-f004:**
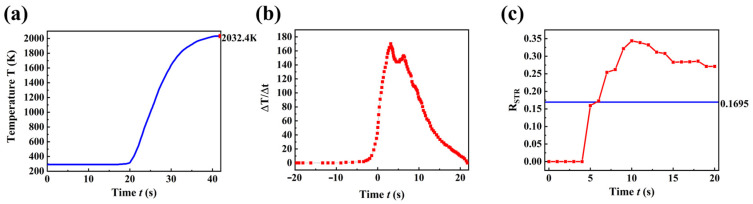
Temporal evolution curves of multiple simulation parameters. (**a**) Photothermal effect simulation results. (**b**) Temporal temperature change curve. (**c**) The ratio parameter R_STR_.

**Figure 5 micromachines-16-00946-f005:**
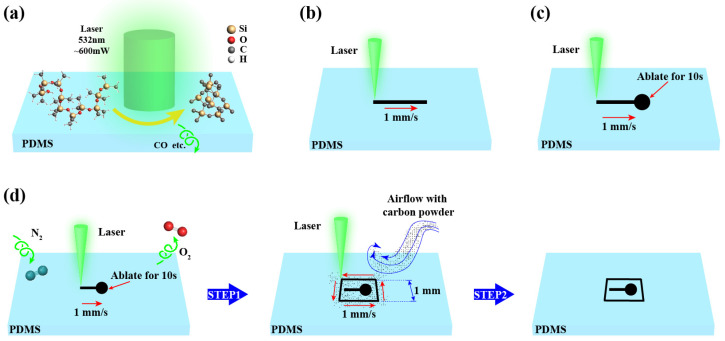
Secondary pyrolysis strategy. (**a**) Laser focused processing. (**b**) Linear machining trajectory. (**c**) Point processing trajectory. (**d**) Frame processing track.

**Figure 6 micromachines-16-00946-f006:**
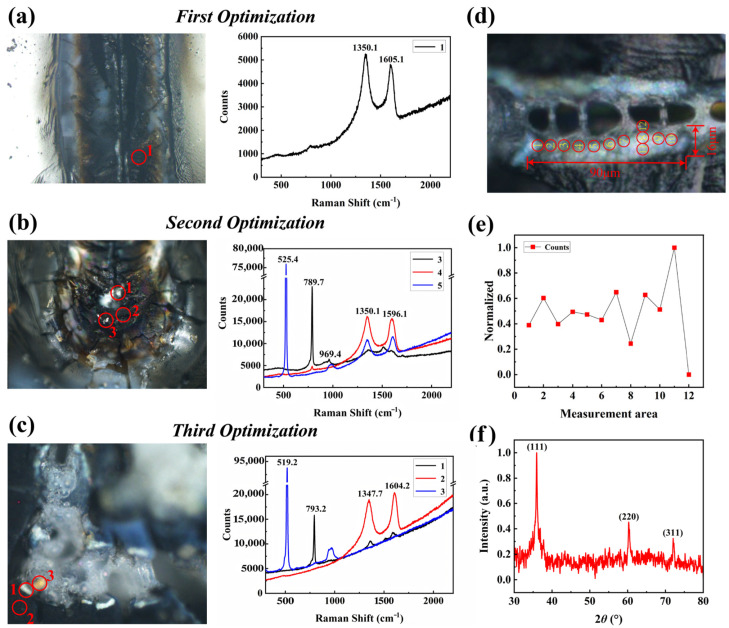
Comparison of multiple processing methods. (**a**) Raman spectroscopy analysis of point ablation regions. (**b**) Raman spectroscopy analysis of line ablation regions. (**c**) Regional Raman spectroscopy analysis after secondary processing. (**d**) Regional distribution of SiC. (**e**) Proportional analysis. (**f**) Intensity analysis.

**Figure 7 micromachines-16-00946-f007:**
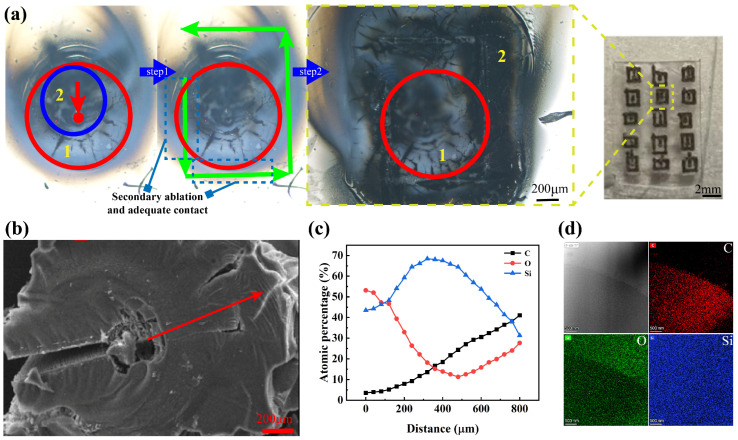
Analysis of secondary pyrolysis processing results. (**a**) Secondary ablation reaction area. (**b**) Observation of the results under a microscope. (**c**) Analysis of changes in composition. (**d**) Regional energy spectral characterization.

## Data Availability

The original contributions presented in this study are included in the article. Further inquiries can be directed to the corresponding author.

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
