# Peer review of "An Efficient Manufacturing Method for Silicon Carbide Crystals in Polymers Based on a Multiscale Simulation-Driven Approach"

_micromachines, 2025, doi:10.3390/mi16080946_

Round 1
Reviewer 1 Report
Comments and Suggestions for Authors
The study presents an innovative multiscale simulation framework to optimize laser pyrolysis of PDMS for high-yield SiC fabrication. While the methodology is rigorous and results are promising, the following revisions are recommended:
- In the manuscript, the precursor formulation (e.g., base polymer type, curing agent ratio like 10:1), curing conditions (temperature, duration, atmosphere), and any substrate pretreatment (e.g., cleaning, surface activation like oxygen plasma) prior to laser processing are not described.
- In general, PDMS is mainly applied in the fields of stretchable electronics/devices, etc. Can the silicon carbide structures prepared in PDMS in the paper be applied in stretchable scenarios?
- Key terms such as "ReaxFF" (reaction force field) and "FWHM" (full width at half maximum) are not defined upon first use (Sections 3.1 and 4.2). Acronyms must be spelled out upon first appearance.
- Equation variables lack clear definitions (e.g., "α(T)" in Equation 1, "I~FTO~" in Equation 4). Symbols should be clearly described upon first use (e.g., "α(T): temperature-dependent absorption coefficient [cm⁻¹]").
- Temperature units are switched between K and °C (e.g., "1400-1600 K" in Section 4.1 vs. "1200-1400°C" in Section 4.2). Key parameters such as heating rate use different formats ("104 K/s" in Section 2 vs. "1014 K/s" in Section 3.2).
Author Response
Response to Reviewer 1 Comments
|
||||||
1. Summary |
|
|
||||
Thank you very much for taking the time to review this manuscript. Please find the detailed responses below and the corresponding revisions changes in the re-submitted files.
|
||||||
2. Questions for General Evaluation |
Reviewer’s Evaluation |
Response and Revisions |
||||
Does the introduction provide sufficient background and include all relevant references? |
Can be improved |
|
||||
Is the research design appropriate? |
Can be improved |
Revised(below for details) |
||||
Are the methods adequately described? |
Can be improved |
Revised(below for details) |
||||
Are the results clearly presented? |
Can be improved |
Revised(below for details) |
||||
Are the conclusions supported by the results? |
Can be improved |
|
||||
Are all figures and tables clear and well-presented? |
Yes |
|
|
|||
3. Point-by-point response to Comments and Suggestions for Authors |
||||||
Comments 1: [In the manuscript, the precursor formulation (e.g., base polymer type, curing agent ratio like 10:1), curing conditions (temperature, duration, atmosphere), and any substrate pretreatment (e.g., cleaning, surface activation like oxygen plasma) prior to laser processing are not described.] |
||||||
Response 1: Thank you for pointing this out. We agree with this comment. Therefore, we have supplemented missing information in the article .This change can be found –page 9 and line298. “[PDMS samples were prepared using Sylgard 184.The base solution and the curing agent were thor-oughly mixed at a mass ratio of 10:1. After mixing, the solution was stirred in a mold for 10 minutes to ensure homogeneous distribution of the components. Since residual bubbles in PDMS during pyrolysis significantly affect its mechanical and thermal properties, degassing was performed after stirring. The specific method involved placing the mixture in a vacuum chamber until complete bubble removal was achieved. Subsequently, the degassed solution was poured into a mold and cured on a hot plate at 80 °C for 30 minutes, completing the preparation of the PDMS substrate.]”
|
||||||
Comments 2: [In general, PDMS is mainly applied in the fields of stretchable electronics/devices, etc. Can the silicon carbide structures prepared in PDMS in the paper be applied in stretchable scenarios?] |
||||||
Response 2: Thank you for pointing this out. We thought about this question and added it in the article. This change can be found – page 12 and line418. “[Simultaneously, the integration of a dispersed array of SiC with the flexible substrate forms a "dispersed-island design" for the brittle material distribution, thereby achiev-ing measurable tensile ductility.]”
|
||||||
Comments 3: [Key terms such as "ReaxFF" (reaction force field) and "FWHM" (full width at half maximum) are not defined upon first use (Sections 3.1 and 4.2). Acronyms must be spelled out upon first appearance.] |
|
|||||
Response 3: Thank you for pointing this out. We agree with this comment. We have supplemented the abbreviation definition.This change can be found – page 4 and line127,page 11 and line357. “[Figure 2. (a) Schematic of the heat transfer module. (b) Flowchart of ReaxFF (reactive force field) simulations. The FTO peak’s FWHM (full-width at half-maximum) decreases from 9.8 cm-1(dot pyrolysis) to 6.2 cm-1, demonstrating a 36.7% improvement in crystallinity.]” |
|
|||||
|
||||||
Comments 4: [Equation variables lack clear definitions (e.g., "α(T)" in Equation 1, "I~FTO~" in Equation 4). Symbols should be clearly described upon first use (e.g., "α(T): temperature-dependent absorption coefficient [cm⁻¹]").] |
|
|||||
Response 4: Thank you for pointing this out. We agree with this comment. Therefore, we have supplemented supplemented information about variables in the formula.This change can be found –page10 and line348. “[IFTO(cm-1) represents the intensity of the FTO peak, IFLO denotes the intensity of the FLO peak, IG is the intensity of the G-band, ID corresponds to the intensity of the D-band, and ISi indicates the intensity of the Si peak. A higher QSiC value indicates better quality SiC.]”
|
|
|||||
Comments 5: [Temperature units are switched between K and °C (e.g., "1400-1600 K" in Section 4.1 vs. "1200-1400°C" in Section 4.2). Key parameters such as heating rate use different formats ("104 K/s" in Section 2 vs. "1014 K/s" in Section 3.2).] |
|
|||||
Response 5: Thank you for pointing this out. We agree with this comment. Therefore, we have corrected the conversion error of the temperature parameter variable. This change can be found – page 7 and line235. “[This asymmetric temperature control strategy accurately reproduces the characteristic rapid heating (~1014 K/s) and gradient cooling features of laser pyrolysis]”
|
|
|||||
|
|

Reviewer 2 Report
Comments and Suggestions for Authors
- Minor spelling etc. mistakes in the text. These are marked on the returned pdf, please consult and correct accordingly.
- We suggest that you remove the References [1] and [2] from the Abstract and relocate them accordingly to suitable points in the Introduction.
- Please add spaces/lines before and after Figures and Tables.
- Please reformat the sections of your paper according to the classic format (1. Introduction, 2. Materials and Methods, 3. Results & Discussion, etc.).
- Especially the 2. Materials and Methods section is absolutely necessary. You will need to state the software that you used (you may include here as subsections the passages where you descrive your modeling parameters etc.). You also need to state your characterization equipment and operational parameters (Raman, SEM, etc.), your testing materials (source of PDMS), laser equipment, etc.
- Please expand on the use of the Beer-Lambert law for modeling laser energy, and how can this correctly model the function of a typical pulsed IR laser used in such laser pyrolysis experiments. See: https://doi.org/10.1016/j.jaap.2016.08.007 and https://doi.org/10.1016/j.apsusc.2019.144096 for additional experimental discussion on the formula linking intensity (laser fluence) and laser operating parameters.

Author Response
Response to Reviewer 2 Comments
|
||||||
1. Summary |
|
|
||||
Thank you very much for taking the time to review this manuscript. Please find the detailed responses below and the corresponding revisions changes in the re-submitted files.
|
||||||
2. Questions for General Evaluation |
Reviewer’s Evaluation |
Response and Revisions |
||||
Does the introduction provide sufficient background and include all relevant references? |
Yes |
|
||||
Is the research design appropriate? |
Yes |
|
||||
Are the methods adequately described? |
Can be improved |
Revised(below for details) |
||||
Are the results clearly presented? |
Must be improved |
Revised(below for details) |
||||
Are the conclusions supported by the results? |
Can be improved |
|
||||
Are all figures and tables clear and well-presented? |
Can be improved |
Revised(below for details) |
|
|||
3. Point-by-point response to Comments and Suggestions for Authors |
||||||
Comments 1: [Minor spelling etc. mistakes in the text. These are marked on the returned pdf, please consult and correct accordingly.]
|
||||||
Response 1: Thank you for pointing this out. We agree with this comment. Therefore, we have corrected the case-sensitive spelling errors in the manuscript. This change can be found – page 3 and line96/105/107/109.
|
||||||
Comments 2: [We suggest that you remove the References [1] and [2] from the Abstract and relocate them accordingly to suitable points in the Introduction.] |
||||||
Response 2: Thank you for pointing this out. We agree with this comment. Therefore, we have removed the References [1] and [2] from the Abstract. This change can be found – page 1 and line14/16.
|
||||||
Comments 3: [Please add spaces/lines before and after Figures and Tables.Please reformat the sections of your paper according to the classic format (1. Introduction, 2. Materials and Methods, 3. Results & Discussion, etc.).] |
|
|||||
Response 3: Thank you for pointing this out. We agree with this comment. Therefore, we have added lines before and after Figures/Tables and reformat the sections of the paper.This change can be found – page 3 and line92/93,page 4 and line126/127 and other Figures/T ables. |
|
|||||
|
||||||
Comments 4: [Especially the 2. Materials and Methods section is absolutely necessary. You will need to state the software that you used (you may include here as subsections the passages where you descrive your modeling parameters etc.). You also need to state your characterization equipment and operational parameters (Raman, SEM, etc.), your testing materials (source of PDMS), laser equipment, etc.] |
|
|||||
Response 4: Thank you for pointing this out. We agree with this comment. Therefore, we have supplemented missing information in the article .This change can be found –page5 and line159, page 7 and line221(the software that we used),page 9 and line298(testing materials) , page 9 and line329(laser equipment) and page10 and line340, page 11 and line367(Raman, SEM, etc.). “[The microscopic reaction process was simulated by the ReaxFF module in Am-sterdam Modeling Suite (AMS) 2020.102. The simulation system employed the NVT ensemble with the Berendsen thermostat algorithm, where the initial ambient temperature was maintained at 298 K, while the maximum pyrolysis temperature was set based on the results of photothermal effect simulations. PDMS samples were prepared using Sylgard 184.The base solution and the curing agent were thor-oughly mixed at a mass ratio of 10:1. After mixing, the solution was stirred in a mold for 10 minutes to ensure homogeneous distribution of the components. Since residual bubbles in PDMS during pyrolysis significantly affect its mechanical and thermal properties, degassing was performed after stirring. The specific method involved placing the mixture in a vacuum chamber until complete bubble removal was achieved. Subsequently, the degassed solution was poured into a mold and cured on a hot plate at 80 °C for 30 minutes, completing the preparation of the PDMS substrate. The specific configuration of the laser direct writing system is as follows: Based on an Olympus BX53 microscope, it comprises a laser source, reflective beam path optics, an objective lens, a motorized positioning system, an inert gas chamber, and a control computer. The system employs a laser wavelength of 532 nm with an output power range of 600–2000 mW. The optical path incorporates a 532 nm reflector (Thorlabs Inc.). The focusing assembly uses a 50× objective lens (Olympus Corporation). Precise sample movement is achieved via a motorized translation stage (Model XWJ-4P, Sanying Pre-cision Control Co.) with a repositioning accuracy of ±1 μm and a maximum travel speed of 5 mm/s. The characterization was performed using a LabRAM Odyssey Raman spectrometer (HORIBA Ltd.). Characterization was performed using a Talos F200X G2 transmission electron micro-scope (Thermo Fisher Scientific Inc.).]”
|
|
|||||
Comments 5: [Please expand on the use of the Beer-Lambert law for modeling laser energy, and how can this correctly model the function of a typical pulsed IR laser used in such laser pyrolysis experiments. See: https://doi.org/10.1016/j.jaap.2016.08.007 and https://doi.org/10.1016/j.apsusc.2019.144096 for additional experimental discussion on the formula linking intensity (laser fluence) and laser operating parameters.] |
|
|||||
Response 5: Thank you for pointing this out. We agree with this comment. Therefore, we have combined with the experimental conditions, the application of the Beer-Lambert method in Equation 1 is explained, and it is understood that the absorption of laser heat by the simulated material needs to take into account a variety of factors.This change can be found – page 4 and line131. “[Given that the geometric dimensions of the PDMS sample (side length: 20 mm, thick-ness: 1 mm) and the laser spot radius (0.1 mm) are significantly larger than the laser wavelength, and considering that PDMS is semi-transparent at the laser wavelength, the laser energy is gradually attenuated with increasing penetration depth, rather than being completely absorbed at the surface. Unlike the simple surface absorption model, the Beer-Lambert law is the standard model describing the exponential attenuation of light within a medium and is applicable to such bulk-absorbing materials. The specific formula is as follows: where z represents the coordinate along the laser propagation direction, and α(T) denotes the temperature-dependent absorption coefficient of the material. During py-rolysis, the chemical structure of PDMS continuously changes (e.g., Si–CH₃ bond breaking, carbon network restructuring), resulting in dynamic evolution of its optical properties (especially the absorption coefficient) with temperature. The original Beer-Lambert law assumes a constant absorption coefficient, whereas this study more accurately describes the actual physical process by introducing a tempera-ture-dependent absorption coefficient α(T).]” |
||||||
|
|
